# Cross-sectional research conducted in the Netherlands to identify relationships among the actual level of patient-centred care, the care gap (ideal vs actual care delivery) and satisfaction with care

Ferogh Mirzad, Jane Murray Cramm, Anna Petra Nieboer

Erasmus School of Health Policy and Management, Erasmus University, Rotterdam, The Netherlands

**Correspondence to**
Jane Murray Cramm;
cramm@eshpm.eur.nl

## ABSTRACT

**Objective** This study aimed to identify relationships among the actual level of patient-centred care (PCC), the care gap (ideal level of PCC vs actual care delivery) and satisfaction with care.

**Design** This study was a cross-sectional survey.

**Setting** This study was conducted at two locations of a Dutch hospital (Nieuwegein and Leidsche Rijn Utrecht).

**Participants** Patients visiting the outpatient clinics for heart failure, chronic obstructive pulmonary disease (COPD) and cancer in March–May 2017 were asked to fill in a questionnaire. Inclusion criteria were diagnosis with COPD, heart failure or cancer and clinic visitation for a regular appointment. A total of 186 patients filled in the questionnaire.

**Primary and secondary outcome measures** Outcomes evaluated were the actual level of PCC, the care gap and satisfaction with care.

**Results** About half (45%) of the respondents were female, 38% had low educational levels and 31% were single. Respondents' mean age was 67.83 ± 10.02 (range, 16–94) years. Patients' experiences with actual care delivery and their conceptualisation of the ideal type of care differed significantly, representing care gaps, in all PCC dimensions. After controlling for background characteristics, patients' experiences with actual delivery and the care gap were related significantly to patients' satisfaction with care (β = 0.17 and β = − 0.41, respectively).

**Conclusions** Patients' experiences with the actual level of PCC and the care gap are important for patients' satisfaction with care.

## INTRODUCTION

Internationally, patient satisfaction is playing an increasingly important role in care-quality reforms and healthcare delivery in general. Satisfied patients are more likely to be compliant and co-operative and to complete treatment regimens. Hence, patient satisfaction has been identified as the way forward to improve health,

### Strengths and limitations of this study

► Given the cross-sectional nature of the study, we were able to assess only relationships among study variables.
► A longitudinal study design is needed to investigate these relationships over time.
► We included only patients with chronic obstructive pulmonary disease, heart failure and cancer.
► We used subjective measures only.

reduce costs and implement reform.[1 2] Hospitals are therefore refocusing healthcare delivery and organisational policies towards patients to improve the quality of care. Research, for example, shows that to improve satisfaction with care among patients, healthcare staff need to respect and respond to patients' choices, needs and preferences and to involve their family members, elements which are at the core of patient-centred care (PCC).[3]

Since the Institute of Medicine (IOM) identified PCC as one of the six dimensions of improving the quality of care,[4] interest in PCC has grown tremendously. In 2001, the IOM defined PCC as: 'Healthcare that establishes a partnership among practitioners, patients, and their families (when appropriate) to ensure that decisions respect patients' wants, needs, and preferences and that patients have the education and support they need to make decisions and participate in their own care'.[5] Although many definitions and models have been developed to describe PCC, the dimensions identified by the Picker Institute have been the most influential. This might be due to the researchers' use of a combination of empirical research and involvement of patients' points of view in developing this model. Their in-depth

BMJ

extensive research, wherein patients were asked to assess PCC, resulted in the identification of the following eight dimensions: 'respect for patient preferences', 'coordination of care', 'information and education', 'physical comfort', 'emotional support', 'involvement of family and friends', 'transition and continuity' and 'access to care'.[3]

A large body of evidence shows that organisations which implement a constellation of interventions aimed at multiple sets of these eight dimensions (eg, training healthcare professionals in PCC consultation (particularly communication and negotiation skills), incorporating patients' individualised needs and expectations into care plans, use of a comprehensive and individualised discharge plan) achieve higher levels of satisfaction with care among their patients.[3] Although previous research has provided insight on the relationships between specific PCC interventions (eg, computerised assessment and goal setting for patients, patient-centred consultation, healthcare provider counselling, specially designed physical environments, hospital discharge planning to secure smooth transitions) and outcomes, we lack insight into patients' experiences in terms of the eight PCC dimensions and how they relate to their satisfaction with care.

Research on satisfaction with care and quality of care indicates clearly that the investigation of patients' experiences with care is not straightforward. Although the quality of care can be measured in multiple ways, patients' experiences are increasingly used to assess the quality of care delivery.[6] Measuring satisfaction with care and experiences with quality of care has, however, been criticised as being unclear in terms of what is actually being measured or determined, and as failing to discriminate effectively between good and bad practice.[7 8] Jenkinson and colleagues,[9] for example, evaluated patient experiences 1 month after hospital discharge and found that even some of the 55% of patients who rated their care as being excellent (assessed in terms of patient satisfaction) cited serious inpatient care problems. Although the patients in their study were generally satisfied with care delivery, some aspects of this delivery did not meet their expectations and led to disappointment. Satisfaction with care was only partially explained by patients' experiences with actual care. Investigation of the discrepancy between expected or ideal care and the actual care received is expected to better explain the relationship between patients' experiences and satisfaction with care.[10] Investigating and truly understanding patient experiences is a highly complex matter which calls for more detailed assessment. Clearly, asking only about patient satisfaction is not enough. Based on the study of Jenkinson and colleagues,[9] we may conclude that satisfaction with care reflects whether care meets a certain standard or norm, whereas examination of actual experiences with care delivery and identification of the aspects of care that do not meet patients' expectations provide a more detailed and fuller picture. To increase our understanding of patient experiences with PCC, we thus need to ask patients about their experiences with actual care, their conceptualisation of ideal care and the gap between them. Given that better experiences with care delivery also result in better patient and organisational outcomes,[6 11] investigation

of the interplay between (disappointing) care experiences and satisfaction with care would be of value. Earlier research among chronically ill adolescents revealed the importance of disentangling experiences in light of patients' experiences with actual care and the care gap (difference between actual and ideal care).[12] Investigating the relationships among satisfaction with care, actual experiences with care and the care gap will provide insight into whether these experiences are actually important, and if so, how important they are. We currently lack such knowledge in the PCC literature. Therefore, this study aimed to identify relationships among the actual level of PCC, the care gap (ideal level of PCC vs actual care delivery) and satisfaction with care.

## METHODS

This study was conducted at two locations of St. Antonius Hospital (Nieuwegein and Leidsche Rijn Utrecht). Patients visiting the outpatient clinics for heart failure, chronic obstructive pulmonary disease (COPD) and cancer were asked by nurses to fill in a questionnaire after visiting their physician during a few days in April 2017. In every outpatient clinic, nurses received the questionnaire from us and were asked to include patients who were eligible for the study. Inclusion criteria were diagnosis with COPD, heart failure, or cancer and visitation of the outpatient clinic for a regular appointment. Diagnoses and reasons for patients' visits were listed on the outpatient clinic schedules and the physicians they were visiting. Patients on these schedules were screened for eligibility, and those who fulfilled the criteria were asked to participate in the study by the nurses. Approximately 240 questionnaires were distributed of which a total of 186 patients actually filled in the questionnaires. Informed consent was obtained from all participants.

### Measures
#### Actual experience with care
Respondents were asked about their actual experiences with the extent to which each of the eight dimensions of PCC was fulfilled. Response categories were 'not at all' (1), 'a bit' (2), 'somewhat' (3), 'very much so' (4) and 'extensively' (5). Higher scores indicated greater occurrence of the PCC dimension. Cronbach's alpha value for this instrument was 0.80, indicating good reliability.

#### Ideal care
Respondents were also asked about their ideal type of care and how important they thought each of the eight dimensions of PCC really was. They rated their level of agreement on a five-point scale ranging from 1 ('not important at all') to 5 ('very important'). Care fitting what patients thought to be important was considered to constitute ideal care. Cronbach's alpha value for this instrument was 0.91, indicating excellent reliability.

#### The care gap
The care gap was assessed by calculating the difference in each respondent's scores for the ideal type of care and

their actual care experience in each PCC dimension. Cronbach's alpha value for these items were 0.86, indicating good reliability.

## Satisfaction with care

Respondents were asked to rate their overall satisfaction with the care provided in the outpatient clinics on a 0–10 scale, where 0 represented the worst hospital possible and 10 represented the best hospital possible. This question was taken from the Hospital Consumer Assessment of Healthcare Providers and Systems (HCAHPS) survey. The Centres for Medicare and Medicaid Services partnered with the Agency for Healthcare Research and Quality, another agency in the federal Department of Health and Human Services, to develop the HCAHPS (www.hcahpsonline.org).

The Medical research Ethics Committee United determined that the rules stipulated in the Dutch Medical Research Involving Human Subjects Act did not apply to this study (file number W17.019) (see http://www.ccmo.nl/en/your-research-does-it-fall-under-the-wmo). Our research did not have a randomised control trial design; participants were not subjected to procedures such as taking a blood sample; the research was not carried out with the intention of contributing to medical knowledge (eg, aetiology, pathogenesis, signs/symptoms and diagnosis) by systematically collecting and analysing data. The main aim of the research was to investigate experiences of participants with care delivery, a process evaluation to improve quality of care delivery, which does not fall under the scope of Medical Research Involving Human Subjects Act (WMO).

## Background variables

Patients were additionally asked about their age, gender, marital status, education level, medication intake (number of medications), comorbidity and which outpatient clinic they visited (for COPD, heart failure or cancer).

## Statistical analyses

First, descriptive statistics were used to characterise the study population, patients' assessments of the eight dimensions of PCC (actual experiences, ideal type of care and the gap between them) and their satisfaction with care in each outpatient clinic. The $\chi^2$ test and analysis of variance were used to detect differences among outpatient clinics. Second, paired-sample t tests were used to investigate differences between ideal and actual care by PCC dimension. Third, we employed correlation analyses to investigate associations among background characteristics, ideal care, actual care, the care gap and satisfaction with care. Fourth, we used a linear regression model to investigate multivariate relationships among background characteristics, actual care, the care gap and satisfaction with care (with listwise deletion of missing cases). Results were considered significant when two-sided p values were ≤0.05. The SPSS software (V.23; IBM Corporation, Armonk, NY, USA) was used for the analyses.

## Patient and public involvement

Patients were not involved in setting up the study design, the recruitment of patients, development of the research questions or outcomes measures.

## RESULTS

Table 1 displays the characteristics of the 186 patients who completed the questionnaire. About half (45%) of the respondents were female, 38% had low educational levels and 31% were single. Respondents' mean age was 67.83±10.02 (range, 16–94) years. These results show no difference in experiences with care (ideal care, actual care and the care gap) or satisfaction with care between heart failure, COPD and cancer patients. Patients with heart failure were older and more of these patients were male compared with patients with COPD and cancer.

Table 2 displays care gap data for the eight PCC dimensions. These results clearly show that patients' experiences

| Table 1 | Descriptive statistics for study participants | | | | |
|---|---|---|---|---|---|
| **Characteristic** | **COPD (n=71)** | **Heart failure (n=50)** | **Cancer (n=65)** | **P value*** | **Total (n=186)** |
| Age (years) | 63.0±13.0 | 69.0±12.2 | 63.2±14.1 | 0.03 | 64.7±13.4 |
| Gender (male) | 45.1% | 66.0% | 50.8% | 0.07 | 52.7% |
| Marital status (married) | 69.0% | 66.0% | 66.2% | 0.92 | 67.2% |
| Education (years) | 12.4±3.9 | 10.7±4.1 | 12.1±3.8 | 0.08 | 11.8±4.0 |
| Comorbidities (no of additional diseases) | 2.0±1.2 | 2.7±1.3 | 1.7±1.0 | <0.001 | 2.1±1.2 |
| Medication (no of medicines taken) | 3.9±1.5 | 4.8±0.7 | 3.3±1.8 | <0.001 | 3.9±1.6 |
| Ideal care | 3.0±0.6 | 3.2±0.5 | 3.1±0.5 | 0.122 | 3.1±0.5 |
| Actual care | 2.3±0.8 | 2.5±0.8 | 2.6±0.8 | 0.238 | 2.4±0.8 |
| Care gap (ideal vs actual care) | 0.6±0.7 | 0.7±0.7 | 0.5±0.8 | 0.592 | 0.6±0.7 |
| Satisfaction with care | 8.1±1.2 | 8.2±1.0 | 8.3±0.8 | 0.501 | 8.2±1.0 |

Data are expressed as mean±SD or percentage.
*Difference among groups, $\chi^2$ test or analysis of variance.
COPD, chronic obstructive pulmonary disease.

**Table 2** Care gaps (actual vs ideal care) experienced by patients in the eight dimensions of patient-centred care

| Dimension | Actual care | Ideal care | Care gap (mean difference) | P value* |
|---|---|---|---|---|
| Patient preferences (n=169) | 2.45±0.94 | 3.04±0.85 | −0.59±0.88 | <0.001 |
| Information and education (n=175) | 2.9±0.92 | 3.60±0.64 | −0.70±0.99 | <0.001 |
| Co-ordination of care (n=175) | 2.69±0.98 | 3.50±0.66 | −0.81±1.11 | <0.001 |
| Emotional support (n=167) | 1.93±1.12 | 2.32±0.99 | −0.38±1.05 | <0.001 |
| Physical comfort (n=167) | 2.49±0.97 | 3.13±0.83 | −0.64±1.05 | <0.001 |
| Family and friends (n=169) | 2.07±1.22 | 2.50±1.06 | −0.44±0.94 | <0.001 |
| Transition of care (n=175) | 2.57±1.07 | 3.21±0.76 | −0.63±1.11 | <0.001 |
| Access to care (n=172) | 2.54±1.05 | 3.17±0.77 | −0.63±1.10 | <0.001 |

Data are expressed as mean±SD.
*Difference between actual and ideal care, paired-sample t test.

with actual care delivery and their views of the ideal type of care differed significantly, revealing care gaps, in all dimensions.

Table 3 shows associations of study variables. Satisfaction with care was related significantly to actual experiences with care (r=0.41, p<0.001) and the care gap (r=−0.37, p<0.001). No significant association was found between any background characteristic and satisfaction with care.

Table 4 shows multivariate relationships among the study variables. After controlling for background characteristics, patients' experiences with actual delivery and the care gap were related significantly to their satisfaction with care (β=0.17 and β=−0.41, respectively). No significant relationship was found between any background characteristic and satisfaction with care.

## DISCUSSION

This study aimed to identify relationships among the actual level of PCC, the care gap (ideal level of PCC vs actual care delivery) and satisfaction with care among patients in three outpatient clinics (for COPD, heart failure and cancer). Importantly, we found that patients' experiences with actual care delivery and their conceptualisation of the ideal type of care differed significantly in all eight PCC dimensions. The study results clearly show that patients experienced care gaps in all of these dimensions, and that their actual experiences and the care gap were related significantly to their satisfaction with care, even after controlling for background characteristics. While we already knew that a constellation of interventions aimed at multiple PCC dimensions resulted in increased patient satisfaction with care,[3] this research additionally shows that patients' (positive and disappointing) experiences with care delivery are associated positively with care satisfaction. Investigation of the care gap especially adds value when actual care is also taken into account. In line with the findings of Sonneveld and colleagues[12] for chronically ill adolescents, this research shows the importance of asking chronically ill adult patients about their experiences with actual care, ideal care and the gap between them to truly understand the relationships of

these experiences to care satisfaction. The identification of PCC dimensions in need of improvement (evidenced by higher gap scores) can be a first step in organisations' efforts to further improve levels of patient-centredness and satisfaction with care. Research clearly shows that poor experiences with access to care are associated with mortality,[13] and that lack of integration among various silos and inadequate communication among providers during transition of care delay the delivery of appropriate healthcare services, leading to poor health outcomes and higher costs.[14]

A noticeable finding are the relatively lower emotional support scores looking at both actual care as well as ideal care. This is in line with the study of Cramm and Nieboer[15] who also investigated these eight dimensions of PCC among multimorbidity patients. In their study, multimorbidity patients also gave the lowest score to the emotional support dimension. Healthcare is known to struggle with achieving real gains in chronically ill patients' emotional or mental well-being because the focus is often mainly aimed at physical health and clinical outcomes only.[16 17] Supporting patients' mental and emotional well-being needs presents a huge challenge in current care delivery[18 19] as this study also indicates.

Another important finding is that no significant associations were found between patients' experiences with care and background variables (age, gender, marital status, type of chronic disease, multimorbidity and medication intake). This is a noticeable finding given that earlier research did show a relationship between background variables (such as age and multimorbidity)[20 21] and experiences with care. Although this finding would suggest that care gaps are pervasive across different specialties (diseases) and demographics, this study is first of its kind and we are therefore cautious to draw such conclusions. Looking at earlier research among multimorbidity patients, we do know that to align with the clinical reality of multimorbidity, care should evolve from a disease orientation to a patient goal orientation, focused on maximising the health goals of individual patients with unique sets of risks, conditions and priorities.[22] The eight

**Table 3** Associations among background characteristics, ideal care, actual care, the care gap and satisfaction with care

| | 1. Age (years) | 2. Gender (male) | 3. Marital status (married) | 4. Education (years) | 5. COPD | 6. Cancer | 7. Heart failure | 8. Comorbidities | 9. Medication | 10. Ideal care | 11. Actual care | 12. Care gap |
|---|---|---|---|---|---|---|---|---|---|---|---|---|
| 1. Age (years) | | | | | | | | | | | | |
| 2. Gender (male) | 0.18** | | | | | | | | | | | |
| 3. Marital status (married) | 0.02 | 0.16* | | | | | | | | | | |
| 4. Education (years) | −0.30*** | −0.02 | 0.14 | | | | | | | | | |
| 5. COPD | −0.10 | −0.12 | 0.03 | 0.11 | | | | | | | | |
| 6. Cancer | −0.08 | −0.03 | −0.02 | 0.05 | −0.58*** | | | | | | | |
| 7. Heart failure | 0.20** | 0.16* | −0.02 | −0.17** | −0.48*** | −0.44*** | | | | | | |
| 8. Comorbidities (no of additional diseases) | 0.30*** | 0.02 | −0.01 | −0.24** | −0.04 | −0.25*** | 0.31*** | | | | | |
| 9. Medication (no of medicines taken) | 0.33*** | 0.11 | 0.06 | −0.20** | 0.00 | −0.31*** | 0.33*** | 0.42*** | | | | |
| 10. Ideal care | 0.13 | −0.22** | −0.05 | −0.00 | −0.15* | 0.06 | 0.10 | 0.12 | 0.12 | | | |
| 11. Actual care | 0.08 | −0.04 | −0.01 | −0.04 | −0.12 | 0.11 | 0.02 | −0.04 | 0.11 | 0.48*** | | |
| 12. Care gap (ideal vs actual care) | 0.02 | −0.14 | −0.03 | 0.05 | 0.01 | −0.07 | 0.06 | 0.11 | −0.04 | 0.22** | −0.75*** | |
| 13. Satisfaction with care | −0.07 | −0.08 | −0.05 | −0.06 | −0.09 | −0.09 | 0.02 | −0.08 | −0.11 | 0.08 | 0.41*** | −0.37*** |

*P≤0.05 (two-tailed); **P<0.01; ***P<0.001.
COPD, chronic obstructive pulmonary disease.

**Table 4** Multivariate relationships between background characteristics, actual care, the care gap (ideal care vs actual care) and satisfaction with care (n=155)

| Independent variable | Satisfaction with care | | |
|---|---|---|---|
| | β | B (SE) | P value |
| Age (years) | −0.07 | −0.01 (0.01) | 0.396 |
| Gender (male) | −0.07 | −0.13 (0.16) | 0.416 |
| Marital status (married) | −0.00 | −0.00 (0.16) | 0.989 |
| Education (years) | −0.07 | −0.02 (0.02) | 0.366 |
| COPD* | −0.13 | −0.27 (0.20) | 0.184 |
| Cancer* | −0.11 | −0.23 (0.22) | 0.291 |
| Comorbidities (no. of additional diseases) | −0.04 | −0.03 (0.07) | 0.644 |
| Medication (no. of medicines taken) | −0.14 | −0.09 (0.06) | 0.123 |
| Actual care | 0.17 | 0.32 (0.15) | 0.036 |
| Care gap (ideal care vs actual care) | −0.41 | −0.57 (0.11) | <0.001 |

*Reference category, heart failure. Results are based on listwise deletion of missing cases. Pairwise deletion and mean substitution of missing cases yielded similar results.
COPD, chronic obstructive pulmonary disease.

dimensions of PCC may be a way to deliver care in such a way that it truly fits the needs and expectations of all patients regardless of their background.

This study has several limitations. First, given its cross-sectional nature, we were able to assess only relationships, not causality. A longitudinal study design is needed to investigate relationships over time. Second, we included only patients with COPD, heart failure and cancer. Given that our findings are in line with those of Sonneveld and colleagues[12] and that the type of chronic disease did not affect patients' experiences or satisfaction with care, we are confident that the inclusion of patients with only three chronic diseases did not influence our study findings. Third, not all eligible patients were systematically asked by nurses to participate in the study. Due to these organisational impediments, we were not able to keep a record of the number of patients who were actually invited to participate by the nurses and who declined.

From this study, we can conclude that patients' experiences with actual care delivery and the care gap are important for patients' satisfaction with care. A deeper understanding of PCC and satisfaction with care thus requires investigation of patients' experiences with actual care, ideal care and the gap between them.

**Correction notice** This article has been corrected since it first published online. The open access licence type has been amended.

**Contributors** FM and APN participated in the initial study design. FM performed data collection. JMC, FM and APN analysed the collected data. JMC and APN drafted the manuscript and contributed to its refinement. All authors read and approved the final version of the manuscript.

**Funding** The authors have not declared a specific grant for this research from any funding agency in the public, commercial or not-for-profit sectors.

**Competing interests** None declared.

**Patient consent for publication** Not required.

**Provenance and peer review** Not commissioned; externally peer reviewed.

**Data sharing statement** No additional data are available.

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
