## [Reviewer comments · BMJ Open]

ARTICLE DETAILS

TITLE (PROVISIONAL)	A cross-sectional research conducted in the Netherlands to identify relationships among the actual level of patient-centred care, the care gap (ideal versus actual care delivery), and satisfaction with care
AUTHORS	Mirzad, Ferogh; Cramm, Jane; Nieboer, Anna

VERSION 1 – REVIEW

REVIEWER	Ann O'Hare University of Washington, USA
REVIEW RETURNED	13-Aug-2018

GENERAL COMMENTS	This is an interesting study that measures the relationship between patient expectations of care, experiences of care and gaps in care using a survey approach. The authors found that the characteristics of individual patients were not associated with any of these measures, but the measures were strongly associated with one another. Most notably, patients for whom there was a less marked gap between actual and expected care were less satisfied with their care. Can the authors spend more time discussing the details of their results, for example expectations for some domains were lower than others, the same was true for expected care. For example, emotional experience of care was ranked lower than some other aspects of care, but patients' expectations were also lower resulting in a less pronounced gap in care. Overall, while the results are not unexpected, I was quite surprised to see that no measured patient characteristics were associated with any of these measures. This actually has large implications for strategies to make care more patient centered and the authors should probably make more of this in the conclusion. Patients expectations and actual experiences of care don't seem to follow any particular pattern based on standard measures such as age, gender, race and types of comorbidity. Can the authors discuss this finding in relation to existing work on patient centered care and patient satisfaction with care? What are the implications of this finding? Does this mean that that gaps in patient care are pervasive across different specialties, different demographic groups? Some of the work around multimorbidity and questioning disease centric models of care may be useful in supporting more discussion around this (see work of Mary Tinetti, Cynthia Boyd and colleagues). A minor point is that it would help to label characteristics on top row of Table 3, it took me a while to understand what the table was showing.
---

REVIEWER	Koula Asimakopoulou Kings College London UK
REVIEW RETURNED	24-Sep-2018

GENERAL COMMENTS	I enjoyed reading your paper - thank you for reporting your work so clearly. I think looking at PCC from the 8 angles that the IOM suggest is a really sound way of ensuring that patients (and healthcare staff) understand what they are rating. I also think your findings are important as they might help healthcare delivery staff design services more appropriately. There are just a couple of things that are missing from the paper; i) ethics approvals- I see you have obtained informed consent but I think you should say a little more about how patients were treated according to guidelines in the study. Did you apply for/ receive ethics clearance? If not, why not? If so, please state who reviewed the study and add a couple of sentences on how you ensured that patients were not coerced into the study, describe their treatment throughout and explain whether you had any drop out or refuse to take part. The second point is related to the one above - please say how many people were eligible, of those how many were approached, of those how many declined / agreed and finally how many ended up participating- finishing with a response rate to your survey. Other than those two minor additions, I think your paper is well-written and sound.
--

VERSION 1 – AUTHOR RESPONSE

Reviewers' Comments to Author:

Reviewer: 1

Reviewer Name: Ann O'Hare

Institution and Country: University of Washington, USA Please state any competing interests or state 'None declared': None

This is an interesting study that measures the relationship between patient expectations of care, experiences of care and gaps in care using a survey approach.

We thank the reviewer for these positive remarks.

The authors found that the characteristics of individual patients were not associated with any of these measures, but the measures were strongly associated with one another. Most notably, patients for whom there was a less marked gap between actual and expected care were less satisfied with their care. Can the authors spend more time discussing the details of their results, for example expectations for some domains were lower than others, the same was true for expected care. For example, emotional experience of care was ranked lower than some other aspects of care, but patients' expectations were also lower resulting in a less pronounced gap in care.

We agree with the reviewer that this is an interesting finding and added the following paragraph to the discussion: A noticeable finding are the relatively lower emotional support scores looking at both actual care as well as ideal care. This is in line with the study of Cramm and Nieboer¹⁵ who also

investigated these eight dimensions of patient-centred care among multimorbidity patients. In their study multimorbidity patients also gave the lowest score to the emotional support dimension. Health care is known to struggle with achieving real gains in chronically ill patients' emotional or mental well-being because the focus is often mainly aimed at physical health and clinical outcomes only.^{16,17} Supporting patients' mental and emotional well-being needs presents a huge challenge in current care delivery^{18,19} as this study also indicates.

Overall, while the results are not unexpected, I was quite surprised to see that no measured patient characteristics were associated with any of these measures. This actually has large implications for strategies to make care more patient centered and the authors should probably make more of this in the conclusion. Patients expectations and actual experiences of care don't seem to follow any particular pattern based on standard measures such as age, gender, race and types of comorbidity. Can the authors discuss this finding in relation to existing work on patient centered care and patient satisfaction with care? What are the implications of this finding? Does this mean that that gaps in patient care are pervasive across different specialties, different demographic groups? Some of the work around multimorbidity and questioning disease centric models of care may be useful in supporting more discussion around this (see work of Mary Tinetti, Cynthia Boyd and colleagues).

We thank the reviewer for pointing out this interesting work and agree with the reviewer that this is an interesting finding and therefore added an extra paragraph to the discussion: Another important finding is that no associations were found between patients' experiences with care and background variables (age, gender, marital status, type of chronic disease, multimorbidity and medication intake). This is a noticeable finding given that earlier research did show a relationship between background variables (such as age and multimorbidity)^{20,21} and experiences with care. Although, this finding would suggest that care gaps are pervasive across different specialties (diseases) and demographics, this study is first of its kind and we are therefore cautious to draw such conclusions. Looking at earlier research among multimorbidity patients we do know that to align with the clinical reality of multimorbidity, care should evolve from a disease orientation to a patient goal orientation, focused on maximizing the health goals of individual patients with unique sets of risks, conditions, and priorities.²² The eight dimensions of PCC may be a way to deliver care in such a way that it truly fits the needs and expectations of all patients regardless of their background.

A minor point is that it would help to label characteristics on top row of Table 3, it took me a while to understand what the table was showing.

We changed the row accordingly.

Reviewer: 2

Reviewer Name: Koula Asimakopoulou

Institution and Country: Kings College London UK Please state any competing interests or state

'None declared': None declared

I enjoyed reading your paper - thank you for reporting your work so clearly. I think looking at PCC from the 8 angles that the IOM suggest is a really sound way of ensuring that patients (and healthcare staff) understand what they are rating. I also think your findings are important as they might help healthcare delivery staff design services more appropriately.

We thank the reviewer for these positive comments.

There are just a couple of things that are missing from the paper; i) ethics approvals- I see you have obtained informed consent but I think you should say a little more about how patients were treated according to guidelines in the study. Did you apply for/ receive ethics clearance? If not, why not? If so,

please state who reviewed the study and add a couple of sentences on how you ensured that patients were not coerced into the study, describe their treatment throughout and explain whether you had any drop out or refuse to take part.

The Medical research Ethics Committee United (MEC-U) determined that the rules stipulated in the Dutch Medical Research Involving Human Subjects Act did not apply to this study (file number W17.019) (see <http://www.ccmo.nl/en/your-research-does-it-fall-under-the-wmo>).

Our research did not have a RCT design, participants were not subjected to procedures such as taking a blood sample, the research was not carried out with the intention of contributing to medical knowledge (e.g. etiology, pathogenesis, signs/symptoms, diagnosis) by systematically collecting and analysing data. The main aim of the research was to investigate experiences of participants with care delivery, a process evaluation to improve quality of care delivery, which does not fall under the scope of Medical Research Involving Human Subjects Act (WMO).

We clarified this in the manuscript. Furthermore, we clarified in the methods section that patients visiting the outpatient clinics for heart failure, chronic obstructive pulmonary disease (COPD), and cancer were asked by nurses to fill in a questionnaire after visiting their physician during a few days in April 2017. Approximately 240 questionnaires were distributed of which a total of 186 patients actually filled in the questionnaires. We added the following text to the manuscript as a limitation: Thirdly, not all eligible patients were systematically asked by nurses to participate in the study. Due to these organizational impediments we were not able to keep a record of the number of patients who were actually invited to participate by the nurses and who declined.

The second point is related to the one above - please say how many people were eligible, of those how many were approached, of those how many declined / agreed and finally how many ended up participating- finishing with a response rate to your survey.

The reviewer correctly noticed this omission. Unfortunately we do not have this information at the individual level. In the methods section we clarified that this study was conducted at two locations of St. Antonius Hospital (Nieuwegein and Leidsche Rijn Utrecht). Patients visiting the outpatient clinics for heart failure, chronic obstructive pulmonary disease (COPD), and cancer were asked to fill in a questionnaire after visiting their physician during a few days in April 2017. In every outpatient clinic nurses received the questionnaire from us and were asked to include patients who were eligible for the study. Inclusion criteria were diagnosis with COPD, heart failure, or cancer and visitation of the outpatient clinic for a regular appointment. Diagnoses and reasons for patients' visits were listed on the outpatient clinic schedules and the physicians they were visiting. Patients on these schedules were screened for eligibility, and those who fulfilled the criteria were asked to participate in the study by the nurses. Approximately 240 questionnaires were distributed of which a total of 186 patients actually filled in the questionnaires. Informed consent was obtained from all participants.

Other than those two minor additions, I think your paper is well-written and sound.